# A Case Study of Enhanced Sulfidization Flotation of Lead Oxide Ore: Influence of Depressants

**Haiyun Xie, Rui Sun, Jizong Wu, Dongxia Feng * and Likun Gao**

Faculty of Land Resource Engineering, Kunming University of Science and Technology, Kunming 650093, China; xie-haiyun@163.com (H.X.); kmust_sun@163.com (R.S.); wjz0799@gmail.com (J.W.); likun_gao@126.com (L.G.)
* Correspondence: dongxia.feng@uqconnect.edu.au; Tel.: +86-0871-65153408

**Abstract:** The refractory lead oxide ore has become an important source of lead metal with the continuous depletion of lead sulfide minerals. Lead oxide ore is of poor floatability and there are few cases to concentrate it successfully. In this study, the sulfidization-xanthate flotation method is applied for the beneficiation of lead oxide ore in Yunnan Province (China) with sodium hexametaphosphate and carboxymethyl cellulose as depressant. Chemical analysis and phase analysis was performed to explore the physicochemical property of raw ore, which provides a research basis for process design and operational control. The main influencing factors during the process, including grinding fineness, reagent types, and dosage, etc., have been studied through flotation tests. Zeta potential measurements and Fourier transfer-infrared spectrometry (FTIR) analysis were conducted to reveal the function mechanism of the reagents. Based on the determined experimental conditions, open circuit tests and closed circuit tests with one stage rougher, three-stage scavenger, and two-stage cleaner flotation, were carried out with the run-of-mine ore with a lead grade of 4.57%. Through close circuit bench test, the lead concentrate with a lead grade of 64.08% and recovery of 92.30% was obtained. This study is of special value, as it provides referencing significance for economically exploiting lead oxide ore.

**Keywords:** lead oxide ore; sulfidization flotation; depressants

---

## 1. Introduction

Lead is an important industry material, which is widely used in electrical, mechanical, chemical, metallurgy, pharmaceutical, military, and other fields [1]. Naturally, lead resources are divided into lead oxide ore and lead sulfide ore. Lead sulfide ore, especially galena, is the main raw material for lead production, because of its good natural floatability [2–4]. However, with the increasing consumption of lead metal, the resources of lead sulfide cannot meet the continuing high demand, so the exploitation and utilization of lead oxide ore is imperative. The lead oxide ore, mainly including cerussite ($PbCO_3$), lead alum ($PbSO_4$), is derived from lead sulfide ore after being interacted with oxygen, carbon dioxide, underground water, and bioorganic matter, etc. The gangue minerals in lead oxide are mainly chlorite, serpentine, calcite, quartz and feldspar, quartz, clay, etc. Thus, very fine particles and a lot of slime are easily generated during crushing and grinding, which greatly interfere with the flotation performance of lead oxide ore [5,6]. In this study, the ore sample from Yunnan Province are cerussite and galena with non-metallic minerals (mainly chlorite and serpentine). The degree of oxidation is quite high, about 80.52%, and the gangue minerals are easily sliming during crushing and grinding. Therefore, the flotation process is deteriorated by a large number of fine particles and the separation process is difficult.

At present, lead oxide ore is mainly enriched by the sulfuration flotation method, while other technologies, such as leaching and roasting, are not widely used because of their high cost [7].

In the case of the lead-oxide ores, sulfurization-xanthates flotation methods are usually applied. After being conditioned by sodium sulfide as sulfurizing reagent, xanthate, and other reagents were added subsequently. The sulfurization-xanthates flotation method has outstanding merits. After being sulfurized by sodium sulphide, the surface of lead oxide ore is modified and the surface electronegativity and hydrophobicity are increased, which is similar to sulfide minerals. Thus the xanthates collector is more easily adsorbed on the mineral surface and forms a lead-xanthates complex, which can significantly improve the floatability of the lead oxide ore without pre-desliming treatment [8]. The sulfidization-xanthate flotation method was successfully employed in many cases, but for different ore deposit, specific research on the equipment operation, process design, and reagents regime are required to obtain a satisfactory result.

Apart from a collector and activator, a depressant plays an important role in controlling the gangue entrainment in fine particle flotation. Typically, sodium silicate and sodium hexametaphosphate are added as depressant in sulfidization-xanthates flotation of oxide ore minerals [9]. In this case, carboxymethyl cellulose (CMC) is tentatively added to help depress fine gangue minerals and improve the flotation performance. CMC is one of important organic macromolecular depressants and it most widely used to depress gangue minerals that contain calcium and magnesium in the flotation process [10–12]. Its possible interaction mechanisms between polymers and the talc surface involve one or more of chemical, electrostatic, hydrogen, and hydrophobic bonding contributions. The generally recognized mechanism is that CMC can adsorb on the gangue mineral surfaces, and the adsorbed hydroxyl groups can form hydration shell through hydrogen bonding, resulting more hydrophilic surface and lower floatability [13–15].

In this work, exploratory research work was focus on the specific raw lead oxide ore with the sulfidization-xanthate flotation method, in aspects of process design and reagents regimes. Chemical analysis and phase analysis was performed to explore the physicochemical property of raw ore, which provides the research basis for process design and operational control. Flotation tests were conducted and results are to diagnose and determine the operation conditions and the flow sheet design. Zeta potential and FTIR characterization were conducted to understand the sulfidization mechanism of sodium sulfide on the lead oxide mineral surface. The depressing mechanism of CMC was studied via zeta potential measurements. This study lays the research foundation for the efficient separation and enrichment of lead oxide resources.

## 2. Materials and Methods

### 2.1. Mineralogical Analysis of Raw Ore

The raw ore is from Tengchong mine, Yunnan Province. Spectrum analysis (ICP-OES, German) was applied to semi-quantitatively determine the element types and their content, as shown in Table 1. Lead, calcium, silicon, magnesium, iron, and aluminum are the main elements in the raw ore. Subsequently, atomic absorption spectrophotometer (4530F, China) was employed to quantify the valuable elements and gangue components, as shown in Table 2. The results showed that the ore has a lead grade of 4.57% and a quite low other metals content. According to the data of element analysis, X-ray diffractometer (XRD, BTX-II, Guangyu Tech, Taipei, Taiwan) analysis was performed to determine the mineralogical composition of the raw ore, as shown in Table 3. The results indicate that the lead oxide ore is mainly sandstone type ore and metallic minerals are mainly cerussite, lead alum and galena, and the gangue minerals are mainly chlorite and serpentine, followed by calcite, quartz, and feldspar, etc., Pb phase analysis was carried out to further confirm the lead mineral type. Lead oxide and lead sulfide contents of the ore were assayed by the titration method and content of insoluble lead was measured by Atomic Absorption Spectrometry (WFX-120, Ruida Tech, Beijing, China). The weight fraction of lead chemical compounds (wB/%) and lead weight fraction in raw ore (wPb/%) were presented in Table 4. It indicates a high degree of oxidation with approximately 80.52% of lead minerals that are distributed in carbonates and sulfate.

**Table 1.** Spectrum Analysis results of raw ore/%.

| Ba | As | Al | S | Mg | Mn | Si | Zn | Fe | Ca | Cu | Pb |
|------|------|------|------|------|------|------|------|------|------|------|------|
| 0.12 | <0.1 | >1 | 0.30 | >1 | <0.1 | >4 | <0.1 | 1 | >5 | <0.1 | >1 |

**Table 2.** Multi-element analysis results of ore/%.

| Pb | Zn | S | Cu | Fe | SiO$_2$ | CaO | MgO | Al$_2$O$_3$ |
|------|------|------|------|------|------|------|------|------|
| 4.57 | 0.01 | 0.33 | 0.01 | 1.21 | 34.56 | 18.24 | 9.31 | 7.65 |

**Table 3.** Mineralogical composition of raw ore/%.

| Cerussite | Lead Alum | Galena | Quartz | Chlorite | Serpentine | Calcite | Feldspar | Others | Total |
|------|------|------|------|------|------|------|------|------|------|
| 3.88 | 0.93 | 0.98 | 10.56 | 29.14 | 28.67 | 12.59 | 8.22 | 5.03 | 100 |

**Table 4.** Analysis results of lead phase.

| Lead Phase | Carbonate | Sulfate | Sulfide | others | Total |
|------|------|------|------|------|------|
| wB/% | 65.76 | 15.76 | 16.61 | 1.87 | 100.00 |
| wPb/% | 3.01 | 0.67 | 0.85 | 0.04 | 4.57 |
| Pb Distribution/% | 65.86 | 14.66 | 18.60 | 0.88 | 100.00 |

### 2.2. Flotation Tests

Rougher flotation tests were carried out by one stage rougher and three stage scavengings to determine the optimum experimental conditions, including the optimal particle size fraction for flotation and the optimum reagents' dosage (Figure 1). The raw ore (300 g) was first ground with a cone ball mill (XMQ-240 × 90). Flotation tests were conducted in a single flotation cell (XFD of 0.75 L and 1 L) with adding sodium sulfide (Na$_2$S) as sulfurizing reagent, carboxymethyl cellulose (CMC), sodium hexametaphosphate (NaPO$_3$)$_6$, and sodium silicate (Na$_2$SiO$_3$) as depressants, butyl xanthate and butyl ammonium aerofloat as collectors, and pine oil as frother. The first step is to determine the optimum size fraction for flotation, i.e., proper grinding fineness. Optimal reagents dosages were determined by flotation tests after confirming the optimum size fraction of the flotation feed. Finally, the open circuit flotation test and closed circuit flotation test was carried out to examine the feasibility of the process. The pH Value during the flotation was monitored by a precise pH meter (FE20, Mettler Toledo, Zurich, Switzerland).

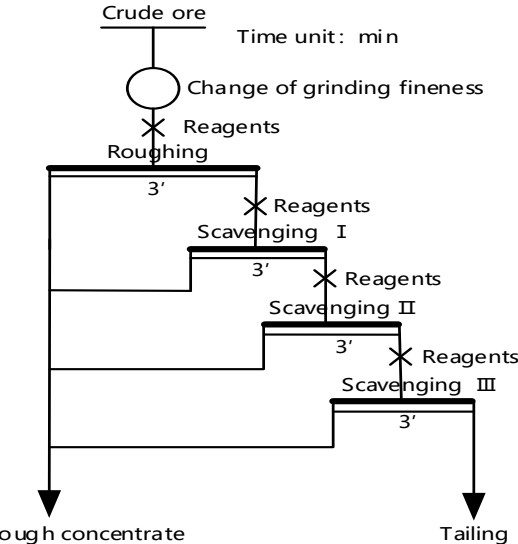

**Figure 1.** Flow chart of tests to determine the grinding fineness and reagents' dosage.

To examine the rheological effect of serpentine on the flotation pulp, rheometer (Anton Paar, MCR 301, Antongpa, Graz, Austria) was used to detect the rheological property of 35% (mass concentration) pulp of Shear stress was measured within shear rate from 50 to 300 s$^{-1}$, as shown in Figure 2. The rheological curves show that the pulp is Newtonian fluid and there is no big difference for the three kinds of samples. It clearly tells that the serpentine in this specific ore has little interference for the pulp viscosity or other rheological properties.

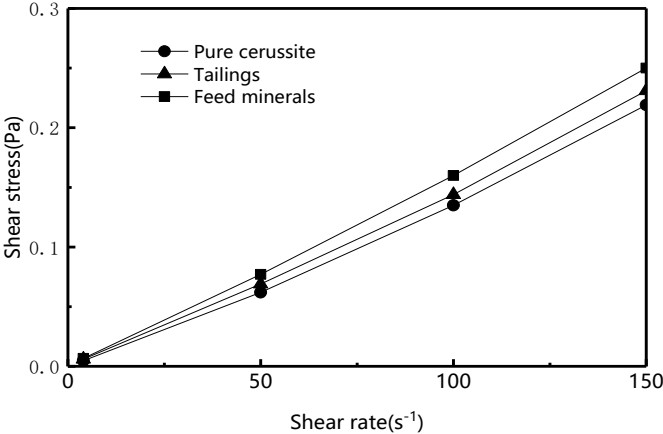

**Figure 2.** Rheological curve of pure cerussite (dots), flotation feed (square) minerals and tailings (triangle) with mass concentration of 35%.

### 2.3. Zeta Potentials Measurements

Zeta potential measurement was conducted by ZetaPlus potential analysis instrument (Brookhaven, NY, USA). The mineral samples were ground in agate mortar until the particle size was less than 2 μm. 20 mg of the mineral samples were conditioned of 0.04% concentration (mass fraction) in a beaker for a desired duration and then added with desired concentrations of the reagents. The pH of the resulting suspension was adjusted by adding 0.1 mol/L sulfuric acid and 0.1 mol/L sodium hydroxide solutions accordingly. The suspension samples were stirred with a magnetic stirrer for 5 min., and let it sit for 5 min. Afterwards, the suspension was injected into the electrode container for measurement. Each measurement was repeated three times and an average of the results was taken as the final result.

### 2.4. FTIR Analysis

Fourier Transform Infrared Spectrometer (IRAffinity-1, Yisai Tech, Shanghai, China) was employed for characterizing the surface chemical properties of the pure galena, pure cerussite, and mineral particles after sulfidization process. According to the literature, the Pb-S absorption peaks appear at wavenumber of 1643.3 cm$^{-1}$ [16], and the characteristic absorption peak for cerussite is approximately 1718 cm$^{-1}$ and 1396 cm$^{-1}$ [17].

## 3. Results and Discussion

### 3.1. Effect of Grinding Fineness

Grinding fineness is an important factor affecting the flotation efficiency of lead oxide ore. The liberation degree would be low if particles were not ground fine enough, and the flotation performance would be negatively affected, both in recovery rate and concentrate grade. However, if the particles were ground too fine, apart from an increase of energy consumption, there would occurs serious sliming which also exerts negative impact on flotation efficiency. Therefore, appropriate grinding fineness should be guaranteed in order to achieve satisfactory results.

Flotation tests process were carried out according to the flow sheet (Figure 1). The slurry concentration in grinding was 65% and the rougher dosage of each reagent was 2000 g/t of sodium

sulphide, 100 g/t of butyl xanthate, 60 g/t of butyl ammonium aerofloat, and 80 g/t of pine oil, with the dosage of reagents for each scavenging was halved in order.

Referring to the grinding tests results that are shown in Table 5, with the increase of the proportion of particles of −0.074 mm, the lead grade and recovery rate increased and then decreased. When the percentage of −0.074 mm particles accounted for 80.00%, the lead grade was 14.01% and recovery rate was 92.49%. Therefore, it is determined that the optimum grinding fineness is the proportion of −0.074 mm, accounting for 80.00%.

**Table 5.** Results of grinding fineness tests.

| −0.074 Proportion/% | Yield/% | Pb Grade/% | Pb Recovery/% |
|---|---|---|---|
| 60.00 | 19.53 | 20.44 | 86.93 |
| 70.00 | 27.77 | 14.09 | 89.33 |
| 80.00 | 28.85 | 14.01 | 92.49 |
| 90.00 | 32.16 | 13.50 | 91.98 |

### 3.2. Effect of Sulfidization Reagent

3.2.1. Determination of Sodium Sulfide Dosage

The aim of suphurization is to change the physical and chemical properties of oxidized mineral surface and modify the ion composition of pulp with adding sulphidiser, so as to enhance the floatability of minerals and achieve the separation and enrichment of oxidized minerals. The right amount of sodium sulfide shows good activation effect for lead oxide ore in sulfidization-xanthate flotation. In solution, sodium sulfide hydrolyses and then releases $S^{2-}$ and $HS^-$ ions into solution. Sulphur ions pass into the crystal lattice of the oxidised minerals from the surface, forming an insoluble subside sulfide surface and enhancing the hydrophobicity of the mineral surface. The $S^{2-}$ can adsorb on surfaces of lead oxide minerals, increasing its hydrophobicity [18]. However, the dosage of sodium sulfide must be strictly controlled, as higher dosage would play a role of depressant and prevent the adsorption of collector on mineral surface [19].

The grinding fineness was −0.074 mm, accounting for 80.00%. The dosage range of sodium sulfide in the Pb rougher flotation was 500–3000 g/t. Other conditions are 100 g/t of butyl xanthate, 60 g/t of butyl ammonium aerofloat, 80 g/t of pine oil, and the dosage for each scavenging was halved in order. As shown in Figure 3, 2000 g/t is the best dosage of sodium sulfide with the lead grade was 16.21% and the recovery was 98.38%.

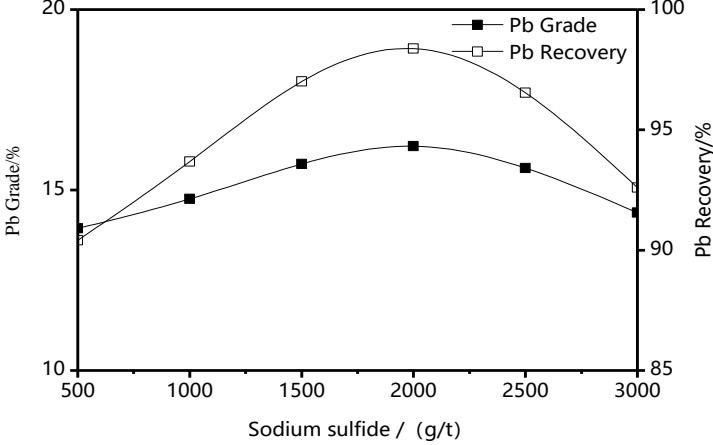

**Figure 3.** The influence of Na₂S dosage on Pb rougher flotation.

### 3.2.2. Characterization of Sulfidization Process

FTIR analysis was conducted with the sulfuring cerussite. The results that are shown in Figure 4 clearly indicate the formation of sulfide on cerussite surface. New absorption peak appears at 1643.3 cm$^{-1}$ when compared with the pure galena and cerussite minerals, which is the characteristic peak of Pb–S. What is more, the peak of cerussite shifts from 1718.16 cm$^{-1}$ to 1722.57 cm$^{-1}$, and the intensity and peak area decreases distinctly, which indicates a reasonable sufidization reaction on cerussite surface.

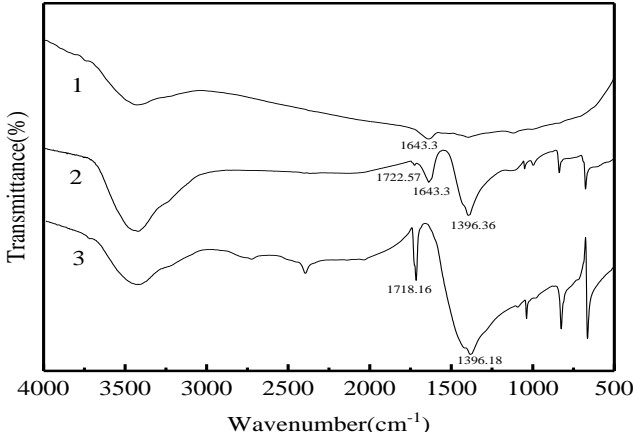

**Figure 4.** Infrared spectra of different samples: (**1**) Pure galena mineral; (**2**) Sulfidized Cerussite; (**3**) Pure cerussite mineral.

The zeta potential of cerussite solution and cerussite with appropriate amount of sodium sulfide (2000 g/t) was measured from pH = 4 to 11. As shown in Figure 5, the cerussite surface was more negatively charged at each condition after adding sodium sulfide. The pH isoelectric point (IEP) of cerussite decreased from 5.85 to 4.82. It indicates the formation of metal sulfide film [20].

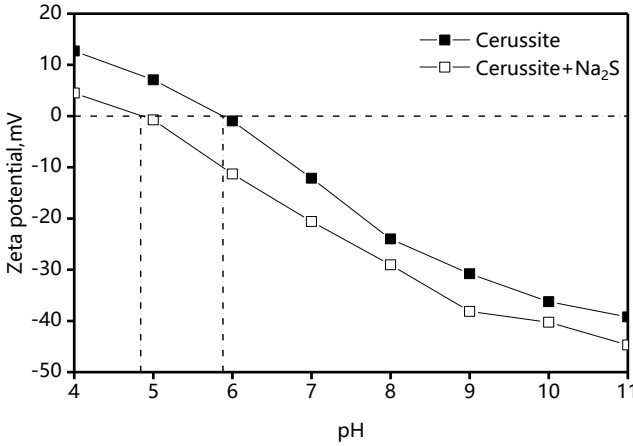

**Figure 5.** Zeta potential of cerussite at different pH in deionized water and sodium sulfide solution.

### 3.3. Effect of Depressing Reagents

#### 3.3.1. Sodium Silicate and Sodium Hexametaphosphate

The high content alkaline gangue in Yunnan lead oxide ore shows serious sliming. Sodium silicate can hydrolyze to silicic acid. $H_2SiO_3$ and $SiO_3^{2-}$ can easily adsorb on the surface of gangue minerals and enhance its hydrophilicity, thus leading to the stable suspension of gangue minerals in solution. However, when the dosage of sodium silicate increases, silicic acid tends to form micelle and further

polymerize and finally precipitate into the solution. Hence, the effective components $H_2SiO_3$ and $SiO_3{}^{2-}$ decrease, which weakens the inhibition ability of sodium silicate on gangue minerals, which results in a decline in the grade and recovery of lead concentrate accordingly [21].

Sodium hexametaphosphate can react with the calcium and magnesium ions in the solution to form a hydrophilic and stable complex, which has a good inhibitory effect on the minerals containing calcium and magnesium [22]. The adsorption of sodium hexametaphosphate on the mineral surface can also increase the electronegativity of the surface and separate the mineral from other negatively charged minerals under the electrostatic repulsion forces. Therefore, sodium silicate and sodium hexametaphosphate were used as depressant (for calcite, dolomite, silicate, etc.) could enhance the slurry dispersion [23]. While its dosage exceeds a modest amount, the complex that was formed by the reaction will not all remain on the surface of calcite, but also disperse in the pulp. After its adsorption onto the lead oxide surfaces, it will also cause inhibition, which results in a decrease in the grade and recovery rate of lead concentrate.

The rougher stage dosages of sodium silicate and sodium hexametaphosphate are 100 g/t + 100 g/t, 200 g/t + 200 g/t, 300 g/t + 300 g/t, 400 g/t + 400 g/t, respectively. The other conditions are grinding fineness −0.074 mm accounting for 80.00%, 2000 g/t of sodium sulphide, 300 g/t of CMC, 100 g/t of butyl xanthate, 60 g/t of butyl ammonium aerofloat, and 80 g/t of pine oil, the dosage for each scavenging was halved in order. As is shown in Figure 6, the lead grade and recovery rate increased up to a certain point and then started to decrease with the increase of the combined dosage of sodium silicate and sodium hexametaphosphate. Therefore, the combined reagent dosage (300 g/t + 300 g/t) is determined as the grade (28.72%) and recovery (95.51%) of lead reach the maximum.

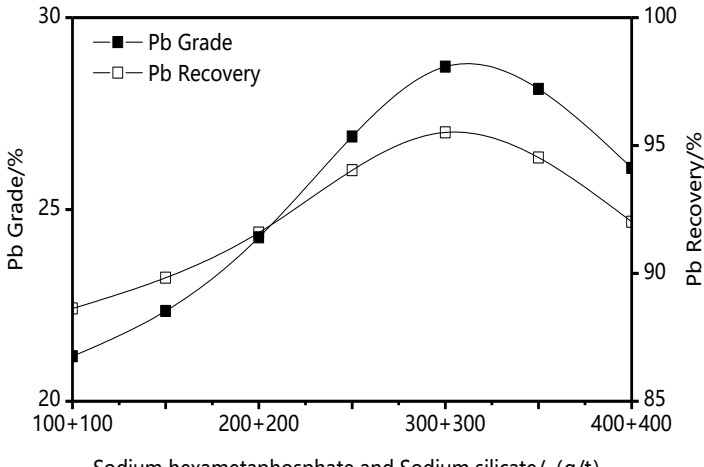

**Figure 6.** The influence of $Na_2SiO_3$ and $(NaPO_3)_6$ dosage on Pb rougher flotation.

### 3.3.2. Carboxymethyl Cellulose (CMC)

The flotation tests were conducted to determine the right dosage of CMC. The grinding fineness was −0.074 mm accounting for 80.00%, the rougher stage dosage range of CMC was from 100 g/t to 400 g/t. For other reagents, it was 2000 g/t of sodium sulphide, 100 g/t of butyl xanthate, 60 g/t of butyl ammonium aerofloat, 80 g/t of pine oil, the dosage for each scavenging was halved in order.

As is shown in Figure 7, the lead grade and recovery rate increased to a certain dosage and then decreased with the increase of CMC dosage. An excessive amount of CMC can increase the electronegativity of not only the gangue surfaces but also the cerussite surfaces, which reduces the adsorption possibility of xanthate collector, resulting in a decrease of the recovery and grade of the cerussite. Therefore, the optimum dosage of CMC was set with 300 g/t with Pb recovery rate of 93.75% and Pb grade of 31.34%.

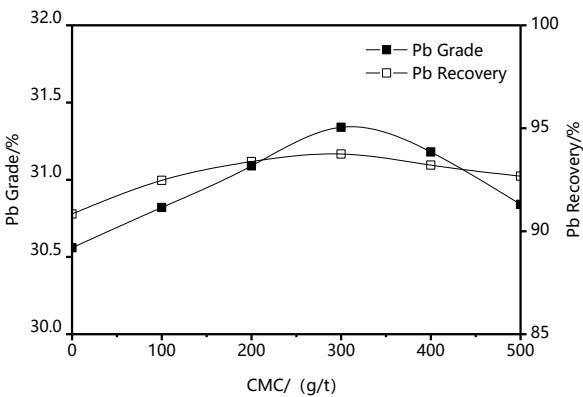

**Figure 7.** The influence of CMC dosage on Pb rougher flotation.

### 3.3.3. Function Mechanism of CMC

As introduced in previous section, the depressing mechanism of CMC has been mostly studied in talc flotation. In our case, the lead oxide ore contains a lot of serpentine and chlorite. It may act as dispersant as well as depressant. CMC is a negatively charged polymer with carboxyl substituent groups along the cellulose chain. It can adsorb on gangue mineral surfaces and alter the surface charge, thus preventing the agglomeration of fine-grained minerals and slime coating on mineral surfaces, hence resulting a good particle dispersion [24,25]. Zeta potential measurement was performed to verify the dispersing mechanism.

Figure 8 shows the zeta potentials of serpentine within different conditions at different pH value. The square line shows the zeta potential of serpentine in deionized water. After adding CMC with concentration of 150 mg/L, the surface potential decreased dramatically. The IEP of serpentine decreases from 9.80 to 5.1. The surface potential was further reduced within the solution of sodium silicate and sodium hexametaphosphate with concentration of 150 mg/L. It went to a quite low value after adding CMC (150 mg/L) as is shown in upside down triangle line. There shows the same trend with chlorite, as shown in Figure 9. The more negative of surface charges, the stronger repulsion occurs, thus preventing the cohesion between the gangue particles. In addition, surface negative electric energy can effectively prevent its interaction with anion collector xanthate.

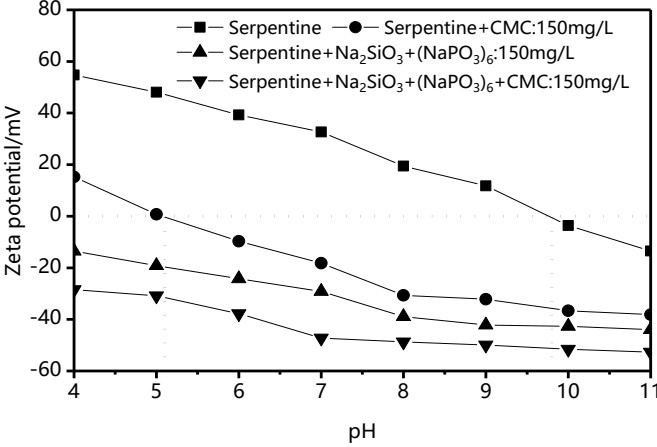

**Figure 8.** Zeta potential of serpentine at different pH.

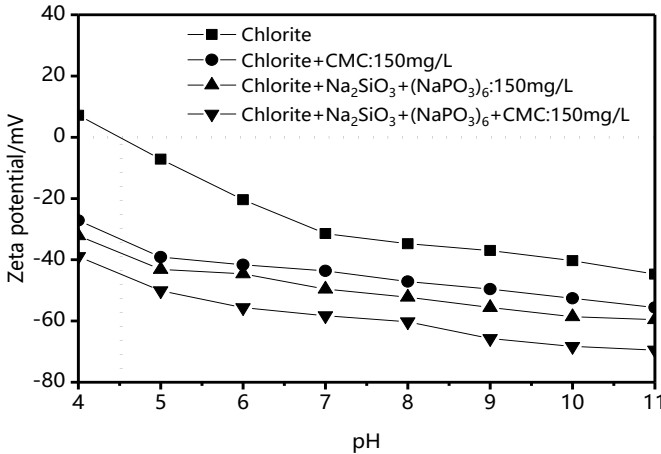

**Figure 9.** Zeta potential of chlorite at different pH.

### 3.4. Effect of Collector Dosage

Xanthate and aerofloat collector molecules dissociate anionic hydrophobic sulfhydryl groups $[(RO)_2CSS]^-$ and $[(RO)_2PSS]^-$ into water. After being sulfurized by sodium sulphide, the surface of lead oxide ore is similar to sulfide mineral PbS. Complexation chemical reaction takes place on mineral surface between $Pb^{2+}$ and the hydrophobic sulfhydryl groups. Stable complexes $[(RO)_2CSS]_2Pb$ and $[(RO)_2PSS]_2Pb$ form on mineral surface through ionic bonding, thus enhancing the hydrophobicity of mineral surfaces. The chemical reactions that may occur on the mineral surface are as following in Equations (1) and (2) [26,27].

$$Pb^{2+} + 2(CH_3CH_2CH_2CH_2O)_2CSS^- \rightarrow 2[(CH_3CH_2CH_2CH_2O)_2CSS]_2Pb \downarrow +2H^+ \tag{1}$$

$$Pb^{2+} + 2(CH_3CH_2CH_2CH_2O)_2PSS^- \rightarrow 2[(CH_3CH_2CH_2CH_2O)_2PSS]_2Pb \downarrow +2H^+ \tag{2}$$

The rougher stage dosages range of butyl xanthate and butyl ammonium aerofloat were between (40 + 40)–(100 + 70) g/t. The other conditions are grinding fineness −0.074 mm accounting for 80.00%, 2000 g/t of sodium sulphide, 300 g/t of CMC, 300 g/t of sodium silicate, 300 g/t of sodium hexametaphosphate and, 80 g/t of pine oil, the dosage for each scavenging was halved in order. Lead recovery reached the peak at the 80 g/t of butyl xanthate and 60 g/t of butyl ammonium aerofloat, while the lead grade and recovery rate decreased at a dosage about 100 g/t + 70 g/t, as shown in Figure 10. As from our perspective of improving the recovery rate, the dosage of 80 g/t + 60 g/t was adopted in the following tests.

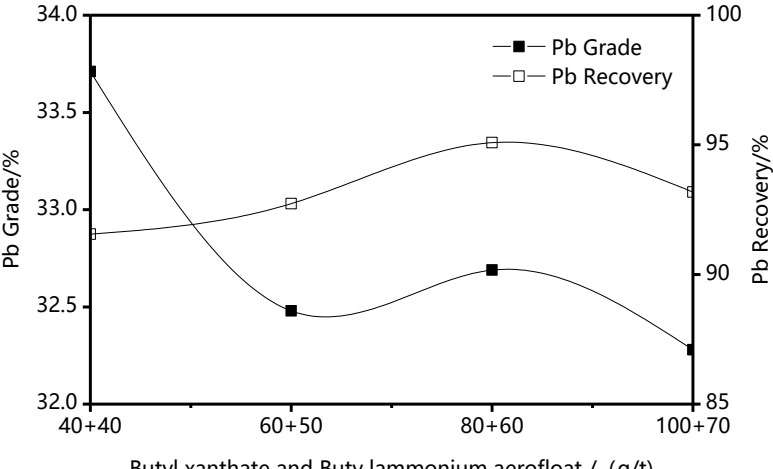

**Figure 10.** The influence of butyl xanthate and butyl ammonium aerofloat dosage on Pb rougher flotation.

### 3.5. Open-Circuit Test

The open circuit tests were performed based on the above results with optimum grinding fineness and reagents' dosage for rougher flotation. The process includes one rougher, two cleaner, and three scavenger stages, as shown is Figure 11. Table 6 gives the reagents dosage for the three stages. The flotation experiments were separately conducted with/without CMC addition for verifying the depressing effect of CMC.

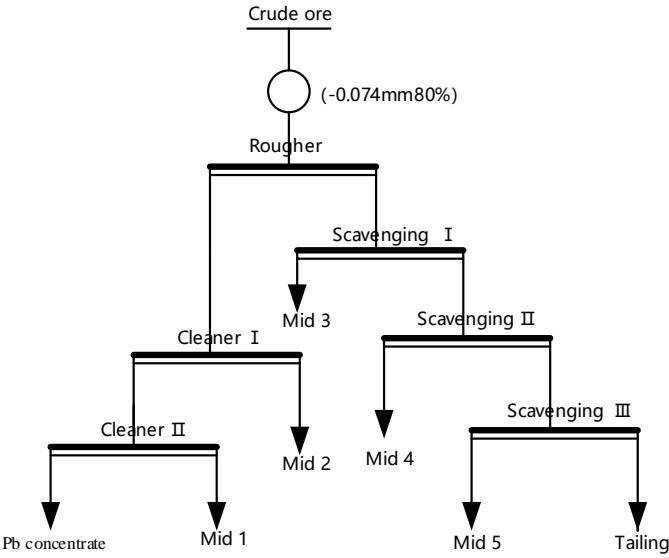

**Figure 11.** Flow-sheets of open circuit test.

**Table 6.** Regent regime of open circuit test.

| Flowsheet | Dosage of Reagents with CMC (g/t) | | | | | | |
|---|---|---|---|---|---|---|---|
| | $Na_2S$ | CMC | $Na_2SiO_3$ | $(NaPO_3)_6$ | $C_4H_6OCSSNa$ | $(C_4H_9O)_2PSSNH_4$ | Pine Oil |
| Rougher | 2000 | 300 | 300 | 300 | 80 | 60 | 80 |
| Scavenger I | 1000 | 150 | 150 | 150 | 40 | 30 | 80 |
| Scavenger II | 500 | 100 | 100 | 100 | 20 | 20 | 80 |
| Scavenger III | 250 | 50 | 50 | 50 | 15 | 10 | 80 |
| Cleaner I | - | - | - | - | 40 | 30 | 80 |
| Cleaner II | - | - | - | - | 30 | 20 | 80 |

Table 7 displays the open circuits test results with one concentrate, one tailing, and five middlings. The lead grade of concentrate can reach as high as 66.59% with lead recovery of 81.92%. The results show that the flotation performance with CMC is much better than that without CMC.

**Table 7.** Results of open circuit test/%.

| Product | CMC | Pb Conc | Midl 1 | Midl 2 | Midl 3 | Midl 4 | Midl 5 | Tailing | Crude Ore |
|---|---|---|---|---|---|---|---|---|---|
| Yield/% | Yes | 5.49 | 1.46 | 5.57 | 3.24 | 1.81 | 3.62 | 78.81 | 100 |
| | - | 5.64 | 1.92 | 6.39 | 3.41 | 2.28 | 3.12 | 77.24 | 100 |
| Pb grade/% | Yes | 66.59 | 10.31 | 1.42 | 6.09 | 2.68 | 1.76 | 0.34 | 4.46 |
| | - | 64.53 | 10.93 | 1.06 | 7.23 | 2.64 | 1.58 | 0.42 | 4.60 |
| Pb recovery% | Yes | 81.92 | 3.37 | 1.77 | 4.42 | 1.09 | 1.43 | 6.00 | 100 |
| | - | 79.16 | 4.56 | 1.47 | 5.36 | 1.31 | 1.07 | 7.06 | 100 |

### 3.6. Closed-Circuit Test

Closed-circuit test was carried out on the basis of the open-circuit test, the five middling products were returned to roughing stage, as shown in Figure 12, to further verify the feasibility and rationality

of the whole process. The addition amount of the reagents was same with that in open circuit test (Table 8).

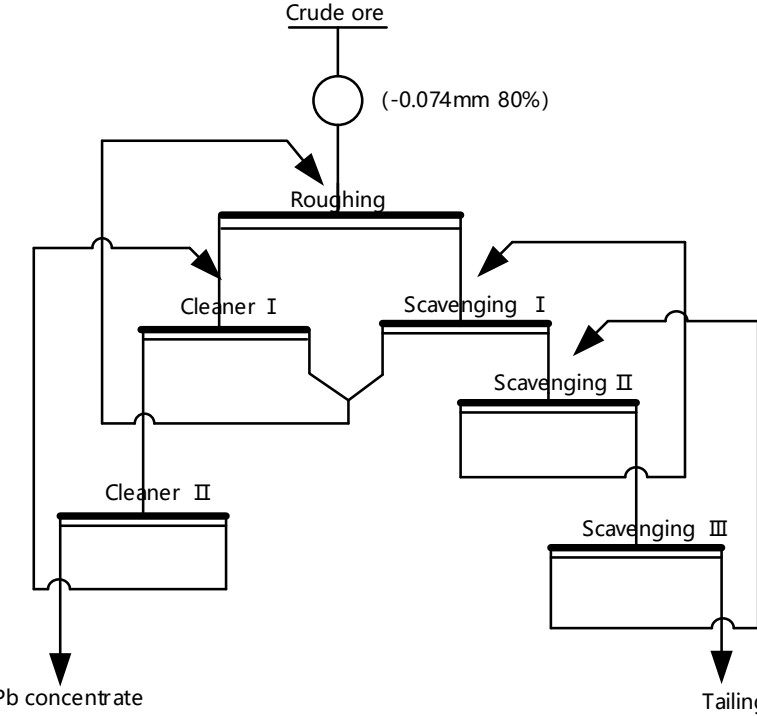

**Figure 12.** Flow-sheets of closed-circuit tests.

**Table 8.** Results of closed-circuit test.

| Product | CMC | Pb Conc | Tailing | Crude Ore |
|---|---|---|---|---|
| Yield/% | Yes | 6.47 | 92.93 | 100.00 |
| | - | 6.59 | 92.84 | 100.00 |
| Pb grade/% | Yes | 64.08 | 0.38 | 4.87 |
| | - | 61.06 | 0.48 | 4.89 |
| Pb recovery% | Yes | 92.30 | 7.26 | 100.00 |
| | - | 90.16 | 9.11 | 100.00 |

When CMC is added throughout the process, the stable closed circuit test results are demonstrated in Table 8, with a satisfactory lead concentrate yield of 6.47%, lead grade of 64.08%, and lead recovery rate of 92.30%, which is significantly improved than that without (grade 61.06% and recovery 90.16%). The results indicate that CMC can significantly enhance the depressing procedure and facilitate the recovery of valuable lead minerals.

## 4. Conclusions

This work was targeted to beneficiate the high-oxidation rate and refractory lead oxide ore in Yunnan Province, China. The reagents regime and flow sheet were appropriate determined. A decent result was obtained from close circuit test with lead concentrate yield of 6.47%, Pb grade of 64.08%, and Pb recovery rate of 92.30%. The addition of CMC can effectively depress the gangue minerals and significantly improve the mineral processing index of lead oxide ore. Adding CMC can increase the grade of lead concentrate by 3.02% and lead recovery by 2.14% when compared with the process without CMC. The sufidization mechanism of sodium sulfide was revealed with zeta potential measurement and FTIR analysis. The function mechanism of CMC was explored with the zeta potential method.

This work displays an integrated research approach for improving the recovering efficiency of the lead oxide ore. However, there is possible room for improving the flotation performance and clarifying the action mechanism of CMC on various gangue minerals. The reagents dosage determination test should consider the interaction effect among the reagents. Additionally, the mechanism and application of other macromolecular reagents in depressing gangue minerals could be further studied.

**Author Contributions:** Conceptualization, H.X. and L.G.; methodology, H.X.; software, J.W.; validation, R.S. and J.W.; formal analysis, J.W.; investigation, R.S.; resources, L.G.; data curation, H.X.; writing—original draft preparation, H.X.; writing—review and editing, D.F.; visualization, R.S.; supervision, D.F.; project administration, H.X.; funding acquisition, H.X. and D.F. All authors have read and agreed to the published version of the manuscript.

**Funding:** This research was funded by the Natural Science Foundation of China, grant number "51464030" and "51804145".

**Acknowledgments:** The authors gratefully acknowledge the academic scholarship of the Chinese Government for the second author.

**Conflicts of Interest:** The authors declare no conflict of interest.

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
