# Peer review of "A Case Study of Enhanced Sulfidization Flotation of Lead Oxide Ore: Influence of Depressants"

_minerals, doi:10.3390/min10020095_

Round 1

Reviewer 1 Report

The reviewer thinks that the paper lacks of the same problems of the original version. Some things were reorganized but no significant improvment is detected. The manuscript is still confusing, discussion is poor, and no rheology effects are mentioned. Based on this this reviewer thinks the paper has to be rejected.

Author Response

Authors appreciate reviewers’ efforts for giving comments and recommendations. We tried our best to revise the manuscript with “track changes” open. Major modifications has been made to logically present the work. Rheological properties have been detected to explain the influence of serpentine on pulp property.

Reviewer 2 Report

The work deals with the flotation of lead oxides by sulfidization using depressants. Although  the subject was studied at the beginning of the past century, this work presents new results which are of value for the flotation industrial processing of lead oxides. The authors showed that the use of depressants (CMC and sodium hexametaphosphate) improved the grade and recovery of Pb. More importantly the authors scientifically explained the role of these depressant in the flotation process. Minor typos: a) Figure 5, Y axis shluld be Pb Grade not Pb graded, b) reference 19 is missing.       

Author Response

    Authors appreciate reviewers’ efforts for giving great comments and recommendations. We tried our best to revise the manuscript with “track changes” open. Please give feedback if there is anywhere required to be revised.

Reviewer 3 Report

The paper presents an important application of sulphidation/sulphidization in oxide mineral flotation. It is generally well-written and clearly presented.

There are several grammatical and paraphrasing issues throughout (e.g., page 5, lines 161-163; page 6, line 168; page 7, line 205-206). Please revise the manuscript and correct all grammatical and punctuation errors.

The main limitation of the study is that only zeta potential measurement was conducted to investigate surface chemistry changes after sulphidation. Zeta potential measurement alone cannot be used to conclude that a metal sulphide has formed on the lead oxide surface (page 6, line 168). The paper can be improved by incorporating additional surface chemistry studies (e.g., XPS, FTIR etc) to confirm the formation of sulphide on the lead oxide ore.

The manuscript could be accepted for publication if the authors are willing to support their findings by advanced surface characterisation techniques.

Author Response

    Authors appreciate reviewers’ efforts for giving great comments and recommendations. We tried our best to revise the manuscript with “track changes” open.  Corrections related to grammatical errors have been made, and the FTIR analysis has been made to verify the formation of the sulfide on lead oxide ore. 

Please give feedback if there is anywhere required to be revised.
